# Methanol Extract of *Dicranopteris linearis* Leaves Attenuate Pain via the Modulation of Opioid/NO-Mediated Pathway

**DOI:** 10.3390/biom10020280

**Published:** 2020-02-12

**Authors:** Zainul Amiruddin Zakaria, Rushduddin Al Jufri Roosli, Najihah Hanisah Marmaya, Maizatul Hasyima Omar, Rusliza Basir, Muhammad Nazrul Somchit

**Affiliations:** 1Department of Biomedical Science, Faculty of Medicine and Health Sciences, Universiti Putra Malaysia, UPM Serdang 43400, Selangor, Malaysia; rush_hour888@yahoo.com (R.A.J.R.); nazrulh@upm.edu.my (M.N.S.); 2Integrative Pharmacogenomics Institute (iPROMISE), Faculty of Pharmacy, Universiti Teknologi MARA, Puncak Alam Campus, Bandar Puncak Alam Selangor 42300, Malaysia; 3Faculty of Business and Management, Universiti Teknologi MARA, Melaka Campus, Melaka 75300, Malaysia; najihah_02@yahoo.com; 4Phytochemistry Unit, Herbal Medicine Research Centre, Institute for Medical Research, Jalan Pahang, Kuala Lumpur 50588, Malaysia; maizatul@imr.gov.my; 5Department of Human Anatomy, Faculty of Medicine and Health Sciences, Universiti Putra Malaysia, UPM Serdang 43400, Selangor, Malaysia; rusliza@upm.edu.my

**Keywords:** *Dicranopteris linearis*, family Gleicheniaceae, antinociceptive activity, mechanisms of antinociception, opioid/NO-dependent pathway, UHPLC/EDI/HRMS analysis, GC-MS analysis, polyphenolics

## Abstract

*Dicranopteris linearis* leaf has been reported to exert antinociceptive activity. The present study elucidates the possible mechanisms of antinociception modulated by the methanol extract of *D. linearis* leaves (MEDL) using various mouse models. The extract (25, 150, and 300 mg/kg) was administered orally to mice for 30 min priot to subjection to the acetic acid-induced writhing-, hot plate- or formalin-test to establish the antinociceptive profile of MEDL. The most effective dose was then used in the elucidation of possible mechanisms of action stage. The extract was also subjected to the phytochemical analyses. The results confirmed that MEDL exerted significant (*p* < 0.05) antinociceptive activity in those pain models as well as the capsaicin-, glutamate-, bradykinin- and phorbol 12-myristate 13-acetate (PMA)-induced paw licking model. Pretreatment with naloxone (a non-selective opioid antagonist) significantly (*p* < 0.05) reversed MEDL effect on thermal nociception. Only l-arginine (a nitric oxide (NO) donor) but not N(ω)-nitro-l-arginine methyl ester (l-NAME; a NO inhibitor) or 1*H*-[1,2,4]oxadiazolo[4,3-a]quinoxalin-1-one (ODQ; a specific soluble guanylyl cyclase inhibitor) significantly (*p* < 0.05) modified MEDL effect on the writhing test. Several polyphenolics and volatile antinociceptive compounds were detected in MEDL. In conclusion, MEDL exerted the opioid/NO-mediated antinociceptive activity, thus, justify *D. linearis* as a potential source for new analgesic agents development.

## 1. Introduction

Pain affects a substantial proportion of the population across the globe and exerts a crucial challenge in public health and clinical medicine [1,2]. The limitations of currently available analgesics such as opioids are mostly because of their adverse effects (i.e., sedation, addiction, nausea, apnea, and constipation) while nonsteroidal anti-inflammatory drugs can produce stroke, myocardial infarction, gastrointestinal ulcers and bleeding [1]. Therefore, initiatives to find safer and potent alternatives are actively conducted and plants have been one of the potential analgesic agents [3]). One of the plants that has been traditionally used to heal pain-related maladies is *Dicranopteris linearis* L., a seedless vascular pteridophytes belonging to the family Gleicheniaceae [4,5].

Scientifically, the chloroform and aqueous extracts of *D. linearis* exert the activity in several animal models namely abdominal constriction test, hot plate test and formalin test [6,7] but no attempt was made to elucidate the possible mechanisms of action involved. Since understanding the mechanisms of antinociception of any potential substance is vital during an attempt to contribute to the drug discovery and development of new analgesics [8] the present study was carried out to establish the antinociceptive activity of the methanol extract of *D. linearis* leaves (MEDL) and to elucidate the possible mechanisms of antinociception involved using various standardized rat or mouse models.

## 2. Materials and Methods

### 2.1. Collection and Identification of Plant

The leaves of *D. linearis* were obtained between February and March 2012 from its natural habitat in Serdang, Selangor, Malaysia based on the previously deposited voucher specimen (SK 2685/15) [9].

### 2.2. Preparation of MEDL

Preparation of MEDL was described in detailed by Zakaria et al. [9]. Briefly, the oven-dried leaves were coarsely grinded and then soaked in methanol (1:20 (*w*/*v*)) for 72 h at room temperature. The supernatant was collected via sequential filtration process using steel filter, cotton wool, and Whatman No. 1 filter papers and the residue was subjected to the same soaking processes for another two times. The supernatants collected from the three filtration processes were pooled together and evaporated (40 °C) under reduced pressure to obtain the crude methanol extract.

### 2.3. Drugs and Chemicals for Plant Extraction and Animal Study

The reagents and drugs used are as listed below: Acetylsalicylic acid (ASA), morphine hydrochloride (MOR), capsaicin, glutamate, bradykinin, phorbol 12-myristate 13-acetate (PMA), naloxone hydrochloride, l-arginine, N(ω)-nitro-l-arginine methyl ester (l-NAME), 1H-[1,2,4]oxadiazolo[4,3-a]quinoxalin-1-one (ODQ) were purchased from Sigma-Aldrich (St. Louis, MO, USA). Formaldehyde was purchased from R & M Chemicals (Essex, England). Acetic acid, dimethyl sulfoxide (DMSO), and methanol were purchased from Fisher Scientific (England). Drugs were dissolved in distilled water. Only ASA and MEDL were prepared by dissolving each of them in 2% DMSO (*v*/*v*) in distilled water. All solutions of drugs, chemicals, and MEDL were administered in the volume of 10 mL/kg.

### 2.4. Reagents and Chemicals for Phytoconstituents Analyses Using the Ultra Performance Liquid Chromatography-Electrospray Ionization-High Resolution Mass Spectrometry (UPLC-ESI-HRMS)

Acetonitrile and formic acid (LC-MS grade) were purchased from Fisher Scientific (M) Sdn. Bhd (Kuala Lumpur, Malaysia), reverse osmosis Milli-Q water (18.2 MΩ) (Millipore, Billerica, MA, USA) was used for all solutions and dilutions. Chemical standards such as gallic acid, catechin, chlorogenic acid, ferulic acid were purchased from Sigma. The standards were diluted in methanol/water, (*v*:*v*, 1:1) to 10 mg/mL and filtered through 0.22 µm membranes prior to LC-MS analysis.

### 2.5. Experimental Animals

Male ICR mice (25–35 g; 5–7 weeks old) were obtained from the Veterinary Animal Unit, Faculty of Veterinary Medicine (FVM), Universiti Putra Malaysia (UPM), Malaysia, and acclimatized for at least 48 h in the Animal Holding Unit, Faculty of Medicine and Health Sciences, UPM (27 ± 2 °C; 70–80% humidity; 12 h light/darkness cycle). Food and water were supplied *ad libitum* [10]. The study protocol of the present study has been approved by the Animal House and Use Committee, Faculty of Medicine and Health Sciences, UPM (Ethical approval no.: UPM/IACUC/AUP–R093/2015). Experiments were conducted between 09:30 and 18:30 h to minimize the effects of environmental changes. The number of animals (*n* = 6) and intensities of noxious stimuli used throughout this study were the minimal necessary to demonstrate the consistent effects of the treatments [11,12].

### 2.6. Evaluation of Antinociceptive Potential of MEDL

#### 2.6.1. Acetic Acid-Induced Abdominal Constriction Test

The acetic acid-induced abdominal constriction test was used to assess the antinociceptive potential of MEDL as described in detailed by Mohd Sani et al. [13]. Briefly, mice (*n* = 6) were treated per os (p.o.) with vehicle (2% DMSO), ASA (100 mg/kg; positive control) or MEDL (25, 150, 300 mg/kg) for 60 min before the administration of phlogistic agent (0.6% acetic acid; intraperitoneal [i.p.]). The total number of abdominal constrictions observed was counted cumulatively for 25 min commencing 5 min after the phlogistic agent administration. Antinociceptive activity was calculated as the percentage inhibition of abdominal constrictions using the following formula: ([mean of control group − mean of test group]/mean of control group) × 100%).

#### 2.6.2. Hot Plate Test

The central antinociceptive potential of MEDL was assessed using the hot plate test according to the procedure described in detailed by Mohd Sani et al. [13]. Briefly, the untreated animals were placed on the hot plate (Model 7280; Ugo Basile, Milan, Italy) heated to 50 ± 0.2 °C to select animals with suitable latency of response (5–7 s) to the thermal-induced nociceptive stimuli. The selected mice (*n* = 6) were pretreated p.o. with vehicle (2% DMSO), MOR (5 mg/kg; positive control) or MEDL (25, 150, 300 mg/kg) for 60 min prior to being subjected to the test. The latency of nociceptive response for all treated groups, recorded before and at 60, 90, 120, 150, 180, and 210 min after the oral administration of the respective test solutions, was statistically compared against the vehicle-treated group.

#### 2.6.3. Formalin-induced Paw Licking Test

The ability of MEDL to affect both the peripheral and central nociceptive mechanisms was further evaluated using the formalin-induced paw licking test as described in detailed by Mohd Sani et al. [13]. Briefly, the mice (*n* = 6) received (p.o.) either vehicle (2% DMSO), ASA (100 mg/kg; standard peripherally-acting analgesic), MOR (5 mg/kg; standard peripherally- and centrally-acting analgesic) or MEDL (25, 150, 300 mg/kg) 60 min prior to the intraplantar (i.pl) injection of 5% (*v*/*v*) formalin (20 µL) into the region of the right hind paw. The latency of discomfort (indicator of pain) specified by the animal spent licking the injected paw was recorded at two phases, namely 0–5 min after formalin injection (known as the early phase) and 15–30 min after formalin injection (known as the late phase).

### 2.7. Determination of the Muscle Relaxant or Sedative Effects of MEDL

To discard the possibility that the extract possesses the non-specific muscle relaxant or sedative effect, which might result in the false positive interpretation of the antinociceptive activity of MEDL, the mice receiving MEDL was also subjected to the rotarod test as described [14]. The apparatus consisted of a horizontal bar with a diameter of 3 cm and subdivided into five compartments (Ugo Basile, model 47600). Twenty-four hours before the experiment, the untreated mice underwent the selection processes wherein each of them was trained on the apparatus by placing them on the rotarod at a fixed speed of 20 rpm and those that were able to remain on the apparatus for 120 sec without falling were selected. The selected mice (*n* = 6) were treated (p.o.) with vehicle (2% DMSO), diazepam (DZP; 4 mg/kg; standard drug) or MEDL (300 mg/kg, p.o.) 60 min before being subjected to the test. The latency took to remain on the apparatus before falling was recorded using a chronometer for 120 s at 5, 10, and 15 min. The average time the mice took to stay on the rotarod equipment for each group was expressed as a result.

### 2.8. Investigation on the Possible Mechanisms of Antinociception of MEDL

#### 2.8.1. Role of Transient Receptor Potential Vanilloid 1 (TRPV1) Receptors

The involvement of TRPV1 receptors in the antinociceptive activity of MEDL was elucidated using the procedure described previously by Lopes et al. [15]. Firstly, mice (*n* = 6) were treated (p.o.) with vehicle (2% DMSO), capsazepine (CAPZ; TRPV1 receptor antagonist; 0.17 mmol/kg) or MEDL (25, 150 or 300 mg/kg), or 60 min before the injection of 20 μL capsaicin (1.6 μg/paw) into the ventral surface of the right hind paw (i.pl) of mice. After the administration of the phlogistic agent, the animals were immediately and individually placed in a glass cage and observed individually for 5 min after the capsaicin injection. The amount of time the mice spent licking or biting the injected paw, which was considered as a nociceptive response, was counted cumulatively with a chronometer and used in the statistical analysis.

#### 2.8.2. Role of Glutamatergic System

Participation of glutamatergic system in the modulation of antinociceptive activity of MEDL was determined according to the previously described method [16]. Mice (*n* = 6) were treated (p.o.) with vehicle (2% DMSO), ASA (100 mg/kg) or MEDL (25, 150 or 300 mg/kg) 60 min prior to the glutamate injection. Then, 20 μL of glutamate (10 μmol/paw) was injected (i.pl) into the ventral surface of the right hind paw of mice. Immediately after the administration of the phlogistic agent, the animals were individually placed in a glass chamber and observed from 0 to 15 min after the glutamate injection. The amount of time the mice spent licking or biting the injected paw, which was considered as a nociceptive response, was cumulatively counted with a chronometer and used in the statistical analysis.

#### 2.8.3. Role of Bradykininergic System

The possible role of bradykininergic system in the mechanisms of antinociception of MEDL was also determined using the method as previously described [17]. Mice (*n* = 6) were orally treated (p.o.) with vehicle (2% DMSO), ASA (100 mg/kg) or MEDL (25, 150 or 300 mg/kg) 60 min before bradykinin injection. A volume of 20 μL of bradykinin (10 nmol/paw) was injected (i.pl) into the ventral surface of the right hind paw of mice. The mice were then observed individually for 10 min following bradykinin injection and the amount of time the mice spent licking the injected paw, which was considered the response of nociception, was recorded using a chronometer and used in the statistical analysis.

#### 2.8.4. Role of Protein Kinase C (PKC)

The involvement of PKC in the antinociceptive effects of MEDL was also studied using a procedure described by Savegnago et al. [18]. Firstly, the mice (*n* = 6) were treated (p.o.) with vehicle (2% DMSO), ASA (100 mg/kg) or MEDL (25, 150 or 300 mg/kg) followed 60 min later by the i.pl injection of 50 μL phorbol 12-myristate 13-acetate (PMA; 0.05 μg/paw; a PKC activator) into the ventral surface of right hind paw of the mice. Mice were observed individually from 15 to 45 min after PMA injection. The amount of time the mice spent licking the injected paw, which was indicative of nociception, was recorded cumulatively using a chronometer and used in the statistical analysis.

#### 2.8.5. Role of Opioidergic System

The involvement of opioid receptors in the antinociceptive activities of MEDL was examined using the protocols described elsewhere [19]. Briefly, the mice (*n* = 6) were pretreated intraperitoneally (i.p.) with 5 mg/kg naloxone, a non-selective opioid receptor antagonist, 15 min before the p.o. administration of vehicle (2% DMSO) or MEDL (300 mg/kg), or i.p. administration of MOR (5 mg/kg). Sixty minutes after the respective test solution administration, the antinociceptive effect was evaluated using the hot plate test as described earlier.

#### 2.8.6. Role of l-Arginine/Nitric Oxide/Cyclic Guanosine Monophosphate (l-arg/NO/cGMP) Pathway

The possible contribution of l-arg/NO/cGMP pathway towards the antinociceptive effect of MEDL was investigated according to the method described by Jiménez-Andrade et al. [20]. Briefly, mice (*n* = 6) were pretreated (i.p.) with 20 mg/kg l-arg (the NO precursor), 20 mg/kg l-NAME (the NO inhibitor), 2 mg/kg ODQ (the specific soluble guanylyl cyclase inhibitor), or their combinations (l-arg + l-NAME or l-arg + ODQ) for 5 min followed by the administration (p.o) of vehicle (2% DMSO), ASA (100 mg/kg) or MEDL (300 mg/kg). Sixty minutes after the administration of test solutions, the mice were subjected to the acetic acid-induced abdominal writhing test as described above.

### 2.9. Phytoconstituents Analyses of MEDL

#### 2.9.1. High-Resolution UPLC-ESI-HRMS Analysis of MEDL

Chromatography separation of MEDL was performed on Dionex Ultimate 3000 RS UHPLC system comprising of a UHPLC pump, an auto sample operating at 4 °C and Exactive Orbitrap mass spectrometer with a heated electrospray ionization probe operating in negative ionization mode (Thermo Fisher Scientific, San Jose, CA, USA). Briefly, reverse separations were carried out using HSS T3 column (2.1 × 50 mm, particle size 1.8 µm; Waters) maintained at 40 °C and eluted at a flow rate of 0.3 mL/min with 25 min gradient of 5–40% of 0.1% acidic acetonitrile in 0.1% aqueous formic acid. The conditions were set as follows: sheath gas at 15 (arbitrary units), aux at 20 and sweep gas at 5 (arbitrary units), spray voltage at 3.0 kV, capillary temperature at 350 °C, and s-lens RF level at 55 V. The mass range was from 100 to 1500 amu with a resolution of 17,000, FTMS AGC target at 2e5, FT-MS/MS AGC target at 1e5, isolation width of 1.5 amu, and max ion injection time of 500 ms and the normalization collision energy at 35% [19].

#### 2.9.2. Gas Chromatography-Mass Spectrometry (GC-MS) Analysis of MEDL

The GC-MS analysis of MEDL was performed using the Agilent 7890A (Agilent Technologies, Palo Alto, CA, USA) coupled with MSD quadrupole detector 5975 C (Agilent Technologies). Separation of analytes by gas chromatography was carried out using a Hewlett Packard HP-5MS silica capillary column (30 m × 0.25 mm × 0.25 mm). For GC-MS detection, an electron ionization system with ionizing energy of 70 eV was used. Helium gas (99.999%) was used as the carrier gas at a constant flow rate of 1 mL/min and an injection volume of 1 μL was employed (split ratio of 1:10), injector temperature was 250 °C, and the ion-source temperature was 280 °C. The oven temperature was programmed from 100 °C (isothermal for 2 min), with an increase of 10 °C /min, to 200 °C and then 12 °C /min to 280 °C, ending with a 17 min isothermal at 280 °C. Mass spectra were taken at 70 eV, a scanning interval of 0.5 sec, and fragments from 45 to 450 Da. Total GC running time was 35.50 min. The relative percentage (%) of the amount of each component was calculated by comparing its average peak area to the total areas and the software used to handle mass spectra and chromatograms was a Turbomass. Interpretation of the obtained mass spectrum GC-MS was conducted using the database of National Institute Standard and Technology (NIST; Version 11.0, Gaithersburg, MD, USA), which have more than 62,000 patterns. The spectrums of the unknown components were compared with the spectrum of the known components stored in the NIST library [19].

## 3. Results

### 3.1. Antinociceptive Profile of MEDL

#### 3.1.1. Effect of MEDL on Nociceptive Response Assessed Using the Abdominal Constriction Test

The antinociceptive potential of MEDL assessed using the acetic acid-induced abdominal constriction test is shown in Figure 1. MEDL at all doses (25, 150 and 300 mg/kg) demonstrated significant (*p* ≤ 0.001) antinociceptive activity with the recorded percentage of antinociception of 42.30, 59.87 and 63.56%, respectively. The antinociceptive effect of 150 and 300 mg/kg MEDL was comparable to 100 mg/kg ASA.

#### 3.1.2. Effect of MEDL on Nociceptive Response Assessed Using the Hot Plate Test

The central antinociceptive potential of MEDL was also assessed against thermal-induced nociception using the hot plate test and is shown in Table 1. MEDL, at the doses of 150 and 300 mg/kg, caused a significant change (*p* ≤ 0.001) in response latency when compared to the vehicle control. However, the 150 mg/kg MEDL only prolonged the latency of nociceptive response a the interval of 60 min while the 300 mg/kg MEDL delayed the nociceptive response latency between the intervals 60 to 120 min. Morphine (MOR), as the standard pain-relieving drug, also exerted significant (*p* < 0.001) antinociceptive activity, which started at the interval of 60 min and prolonged until the end of the experiment (interval of 210 min).

#### 3.1.3. Effect of MEDL on Nociceptive Response Assessed Using the Formalin-induced Paw Licking Test

The antinociceptive potential of MEDL against inflammatory- or non-inflammatory-mediated nociception was also evaluated using the formalin-induced paw licking test and the results are shown in Figure 2. MEDL was found to significantly (*p* ≤ 0.01) reduce the amount of time taken to respond to the nociceptive stimuli in both the early and late phases of formalin-induced paw licking test when compared to the untreated group. In the early phase, only 150 and 300 mg/kg MEDL caused a significant (*p* ≤ 0.01 and *p* ≤ 0.001) reduction in the latency of response to nociception with the percentage of antinociception recorded in the range of 3–45% (Figure 2A). Conversely, in the late phase, all doses of MEDL significantly (*p* ≤ 0.01, *p* ≤ 0.001 and *p* ≤ 0.001, respectively) and dose-dependently reduced the latency of response to nociception with the percentage of antinociception recorded in the range of 23–45% (Figure 2B). Comparatively, ASA, a peripherally-acting antinociceptive drug, significantly (*p* ≤ 0.001) inhibited only the latency of second phase nociception whereas MOR, a centrally- and peripherally-acting antinociceptive drug, significantly (*p* ≤ 0.001) reduced the latency of both phases of nociception with the recorded percentage of antinociception that is higher than MEDL in both phases of nociception when compared to the untreated group. 

### 3.2. Effect of MEDL on Motor Coordination Assessed Using the Rotarod Test

Effect of MEDL on the motor coordination of mice was also measured using the rotarod test and the results obtained were presented in Figure 3. At the dose of 300 mg/kg, MEDL (p.o.) did not cause any significant (*p* > 0.05) change on the motor coordination of mice when compared to the vehicle-treated group. However, diazepam (DZP; 4 mg/kg; p.o.; standard drug) significantly (*p* > 0.001) reduced the time spent by the animals on the revolving rod.

### 3.3. Possible Mechanisms of Antinociception Exhibited by MEDL

#### 3.3.1. Involvement of TRPV1 Receptors in the Modulation of MEDL-induced Antinociceptive Activity

The antinociceptive effect of MEDL against capsaicin-induced paw licking test is shown in Figure 4. All doses of MEDL demonstrated a significant (*p* < 0.05) attenuation of capsaicin-induced nociception with the inhibition ranging between 19–33%.

#### 3.3.2. Involvement of the Glutamatergic System in the Modulation of MEDL-induced Antinociceptive Activity

The antinociceptive effect of MEDL against glutamate-induced paw licking test is shown in Figure 5. All doses of MEDL caused a significant (*p* < 0.05) decrease of glutamate-induce nociception with the inhibition ranging between 15–45%.

#### 3.3.3. Involvement of the Bradykininergic System in the Modulation of MEDL-induced Antinociceptive Activity

All doses of MEDL exhibited significant (*p* < 0.05) inhibitory effect against the nociceptive response induced by bradykinin in a dose-independent manner (Figure 6). The antinociception recorded was in the range of 34–76%.

#### 3.3.4. Involvement of PKC Pathway in the Modulation of MEDL-Induced Antinociceptive Activity

MEDL was also found to exert significant (*p* < 0.05) inhibition of PMA-induced nociception. However, this activity was observed only at the dose of 300 mg/kg with the recorded antinociception ranging between 5–36%. In comparison, 100 mg/kg ASA recorded the antinociception of approximately 54% (Figure 7).

#### 3.3.5. Involvement of the Opioidergic System in the Modulation of MEDL-induced Antinociceptive Activity

Table 2 showed the effect of opioid receptors inhibition on the antinociceptive activity of MEDL assessed using the hot plate test. The antinociceptive activity of MEDL, at the dose of 300 mg/kg, was earlier observed between the 60 and 120 min intervals. Pre-treatment with 5 mg/kg naloxone significantly reversed (*p* < 0.001) the antinociceptive effect of 300 mg/kg MEDL. Similarly, naloxone also significantly (*p* < 0.001) reversed the antinociceptive effect of 5 mg/kg MOR until the end of the experiment. 

#### 3.3.6. Involvement of the l-arginine/NO/cGMP Pathway in the Modulation of MEDL-induced Antinociceptive Activity

The effects of l-arginine, l-NAME or their combination on MEDL antinociception was assessed using the abdominal constriction test and the findings are shown in Figure 8. l-Arginine alone did not affect the acetic acid-induced nociception but significantly (*p* ≤ 0.01) reversed the MEDL antinociceptive activity. On the contrary, l-NAME alone exerted significant (*p* ≤ 0.001) antinociceptive activity and maintained the MEDL-induced antinociception as seen when the extract was given alone. l-arginine was also found to reverse the l-NAME-induced antinociception but when these two compounds were combined and given together with MEDL, the extract’s antinociceptive activity was significantly (*p* ≤ 0.01) reversed but not complete inhibited.

Further investigation has shown that ODQ alone exerted significant (*p* ≤ 0.001) antinociceptive activity but when given together with 300 mg/kg MEDL failed to affect the extract antinociceptive activity (Figure 9). Furthermore, l-arginine failed to reverse ODQ’s antinociceptive activity and their combination (l-arginine + ODQ) also failed to affect MEDL’s antinociceptive activity.

### 3.4. Phytoconstituents of MEDL

#### 3.4.1. UHPLC-ESI-HRMS Profile of MEDL

Chemical constituents of MEDL were analyzed by reversed phase UHPLC-ESI-HRMS, using a gradient mobile phase consisting acetonitrile and aqueous formic acid that allowed for a comprehensive elution of plant analytes i.e., flavanols and flavones and hydroxycinnamic within 20 min. Thoroughness in identification was due in part to a higher sensitivity of the UHPLC-MS and processing Xcalibur software. Metabolite assignments were made by comparing retention time, MS data (accurate mass, isotopic distribution and fragmentation pattern in negative ion mode) of the compounds detected with compounds detected in the literature and database. Identification was confirmed with standard compounds whenever available in-house. The identities, retention times, and observed molecular ion for individual components are presented in Table 3 with a total of 30 metabolites identified (Figure 10; Table 3). Some of the important phytoconstituents identified in MEDL were gallic acid, ferulic acid, protocatechuic acid, caffeic acid, p-coumaric acid, rutin, isoquercitrin, astragalin, catechin, quercetin, apigenin, and kaempferol.

#### 3.4.2. GC-MS Profile of MEDL

Figure 11 shows the GC-MS chromatogram profile of MEDL with a total of 48 peaks detected. Of these, 7 major peaks were identified as follows: (**1**) triphenylphosphine oxide (17.52%), (**2**) 9,12,15-octadecatrienoic acid (13.43%), (**3**) hexadecanoic acid (9.70%), (**4**) tri(2-ethylhexyl) trimellitate (7.98%), (**5**) erucylamide (5.45%), (**6**) 5,10-dihexyl-5,10-diihydroindolo[3,2-b]indole-2,7-dicarbaldehyde (4.63%) and (**7**) linoleic acid (4.17%).

## 4. Discussion

We have earlier reported on the ability of chloroform (CEDL) and aqueous (AEDL) extract of *D. linearis* leaf to exert peripherally- and centrally-mediated antinociceptive activity [6,7]. These findings confirmed that both the lipid-soluble/non-polar and water-soluble/polar bioactive compounds present in *D. linearis* leaf possess the ability to attenuate nociceptive response. It is interesting to highlight that the phytochemical screening of AEDL, CEDL and MEDL revealed that the methanol extract, which contains flavonoids, tannins and saponins, possesses the highest total phenolic content and the most remarkable antioxidant activity [21]. Taking these findings into consideration with the aim of extracting the maximum amount of antinociceptive-bearing bioactive compounds from *D. linearis* regardless of their polarity/solubility, methanol extraction was, therefore, prepared and used in the present study.

The results of the present investigation revealed that MEDL successfully attenuates nociceptive response when assessed using the acetic acid-induced abdominal constriction test, hot plate test, and formalin-induced paw licking test. It is well known that intraperitoneal injection of acetic acid triggers inflammation by casing the resident peritoneal macrophages and mast cells to release several inflammatory mediators (i.e., TNF-α, IL-1β, IL-8, bradykinin, substance P, serotonin and histamine) into the peritoneal cavity, which, in turn, stimulate peripheral nociceptive neurones [22,23]. Thus, the ability of MEDL to attenuate nociception in the abdominal constriction test suggests the extract potential to attenuate inflammatory-mediated peripheral nociceptive response as seen with ASA, a standard peripherally-acting analgesic [22,24]. However, being a classic non-selective nociceptive model, drugs such as muscle relaxants and adrenergic receptor agonists can also reduce abdominal constriction, which could lead to a false positive conclusion [25]. Hence, additional investigations using other nociceptive models such as the hot plate test and formalin-induced paw licking test are essential.

The hot plate test, which involves exposure of rodents onto a plate heated to a constant temperature, generates two behavioural parts, namely paw licking and jumping, as a result of the reflex latency reactions to thermal stimulation of non-inflamed paws. According to Lavich et al. [26] both reactions are considered to be a centrally (supraspinally) integrated responses and, therefore, are sensitive to centrally-acting, but not peripherally-acting analgesics. The ability of MEDL to prolong the latency of discomfort towards thermal thermal-induced nociceptive response indicates the extract potential to attenuate centrally-mediated nociception, a characteristic of the standard analgesic such as MOR. Moreover, since the hot plate test does not involve any motor response modulation or the process of inflammation, but is solely a supraspinal reflex, it is worth to mention on the ability of MEDL to attenuate non-inflammatory mediated nociceptive response and confirmed that the extract possessed antinociceptive activity.

Further study using the formalin-induced paw licking test (formalin test) could also provide information on the ability of MEDL to affect the peripheral and/or central level of nociception. Formalin injection into the i.pl region of the mice’ hind paw triggers two distinct phases of nociceptive behaviour consisting of licking, tonic flexion and phasic flexion (paw jerk) of the injected limb that are described as the early/first and late/second phases [27]. The early phase, which is a result of formalin direct action on nociceptors, is observed instantaneously after the formalin injection and persists for 5 min while the late phase, which results from the generation of inflammatory processes in the periphery and spinal cord and activation of the neurons in the dorsal horns of the spinal cord, emerges between 15 and 60 min following the formalin injection [27]. Due to the different in mechanisms seen both phases, the early phase is also called a neurogenic pain or a non-inflammatory-mediated pain whereas the late phase is also known as an inflammatory-mediated pain due to the tonic response resulted from the release of inflammatory mediators [14]. It is well acknowledged that the centrally acting analgesics inhibit both phases of formalin test whereas the peripherally acting analgesics attenuate only the late phase of formalin test [14]. The ability of MEDL to attenuate both phases of nociceptive responses in the formalin test further suggests that the extract possessed a characteristic of the centrally acting analgesics and might possibly interact with opioid receptors to attenuate nociceptive response.

Since MEDL was proven to act at central level, further investigation needs to be performed to assess the extract’s effect on motor coordination, balance and equilibrium ability of the animals [28]. The rotarod test is frequently used to determine the possible disturbance in the motor coordination by centrally acting compounds/extracts as it is important to make sure that this class of compounds/extracts did not cause depression of the CNS when used as an antinociceptive agent. The results revealed that MEDL did not interfere with the motor coordination of treated animals.

Following the establishment of antinociceptive profile of MEDL, attempt was also made to elucidate the possible mechanisms of antinociception modulated by MEDL with the role of TRPVI, glutamatergic, bradykininergic and opioidergic systems, PKC, and NO/cGMP channels pathways investigated. Since these systems play a crucial role in the modulation of nociceptive response, they are regarded as novel and promising drug targets for the treatment of inflammatory- and non-inflammatory-mediated pain in humans. The role of TRPVI receptors in the modulation of MEDL-induced antinociceptive effect was evaluated using the capsaicin-induced paw licking test. Capsaicin activates the primary afferent fibres through the stimulation of TRPV1 receptors found in the PNS and CNS [29]. The activation of these receptors leads to a series of complex processes in the PNS and transmits nociceptive information to the spinal cord. Interestingly, the TRPVI receptors upregulation or sensitivity is influenced among others by the presence of pro-inflammatory mediators such as bradykinin, phosphorylation mediated by PKC or inactivation of opioid receptors [30]. Interestingly, Vetter et al. have demonstrated that MOR, an opioid receptor agonist, acts peripherally via inhibition of adenylate cyclase to inhibit PKA-potentiated TRPV1 responses, thus, confirmed the association between peripheral opioid receptors and TRPV1 responses in inflammation [31]. In addition, the TRPV1 receptors have also been reported to co-express with many other receptors (such as histamine 1 (H1) receptors, purine (P2) receptors, acid-sensing ion channels, interleukin (IL-1) receptors, and prostaglandin E2 (PGE2) receptors) that are also activated by chemokines and cytokines at sensory terminals [29,32]. Although the present study showed that MEDL inhibited capsaicin-induced nociceptive behaviour, thus, suggesting that the extract may partly interacts with TRPV1 receptors and inhibit it, the selective blockage of TRPV1 receptors by MEDL may not be sufficient to completely attenuate capsaicin-induced nociceptive response due to the complex interaction between the receptor with various nociceptive-modulated factors as described above.

The role of glutamatergic system in the modulation of antinociceptive activity exerted by MEDL was evaluated using the glutamate-induced paw licking test. Evidence from the previous studies suggests that glutamate plays an important role in nociceptive processing since this excitatory amino acid and its receptors are located in areas of the brain, spinal cord, and periphery that are involved in pain sensation and transmission [33]. Glutamate acts at several types of receptors, including ionotropic (such as *N*-methyl-d-aspartate (NMDA) receptors) and metabotropic receptors, as well as the opioid receptor system. NMDA receptors, in part, are found on myelinated and non-myelinated primary afferents at both the peripheral and central terminals. Central activation of presynaptic NMDA receptor by glutamate binding increases calcium influx leading to increase release of glutamate, which further contributes to the central sensitization via the subsequent increase in postsynaptic membrane excitation. Influx of calcium ions mediated partly by NMDA receptors is known to activate, among others, PKC to phosphorylate and sensitizes the TRPV1 receptors leading to the nociceptive response [30]. In comparison, the peripherally-acting NMDA receptors are essential to peripheral sensitization in inflammation with expression amplified and reversal of hyperalgesia and spontaneous pain behaviour with peripheral infiltration of NMDA antagonists. The presence of these receptors might partly explain why the application of glutamate to the spinal cord or periphery induces nociceptive behaviours while inhibition of glutamate release, or of glutamate receptors, in the spinal cord or periphery attenuates both acute and chronic pain in animal models. Interestingly, Rodríguez-Muñoz et al. [34] have further reported on the association between opioid receptors, particularly the mu-receptors, and NMDA receptor subunit in the postsynaptic structures of PAG neurons. They also reported that MOR disrupts this complex by PKC-mediated phosphorylation of the NMDA subunits leading to MOR tolerance whereas the inhibition of PKC activity preserves the complex association and maintain the analgesic effect of MOR. These opposing actions of the MOR and NMDA receptors in pain control could, therefore, be exploited in developing bifunctional drugs that would act exclusively on those NMDA receptors associated with MORs [34]. The present study revealed the ability of MEDL to inhibit glutamate-induced nociceptive response indicating the involvement of glutamatergic system in the antinociceptive activity MEDL. The fact that activation of TRPV1 receptors leads partly to the liberation of glutamate that contributes to nociceptive transmission [29,33] seems to further support the present findings in which MEDL either inhibits glutamate action directly or indirectly via the inhibition of TRPVI receptors leading to the observed antinociceptive activity.

The involvement of bradykininergic system in the modulation of MEDL-exerted antinociceptive activity was analysed using the bradykinin-induced paw licking test. Bradykinins, a small proinflammatory peptide released during tissue damage and inflammation, is involved in the initiation of pain in the periphery and in the development of hypersensitivity in inflamed or injured tissues [35]. In normal tissue bradykinin causes an acute sensation of pain by an action at B_2_ receptors but in inflamed tissue the pharmacology of the response changes to that of B_1_ receptors [35,36]. Other than sensitizing nociceptors, bradykinin was also cited to enhance glutamatergic synaptic transmission between spinal cord neurons and to sensitize TRPV1 in dissociated DRG neuronal cultures partly via the activation of PKC produced downstream to B_2_ receptor activation [37]. Meanwhile, the activation of B_2_ receptors predominantly resulted in the activation of a series of biochemical changes leading to Ca^2+^ release, which helps to prolong the activation of PKC pathway resulting in direct phosphorylation of TRPVI receptors. Increased gain of sensory input by TRPV1-induced enhancement of glutamate release and its potentiation by various inflammatory mediators may contribute to increase nociceptive behaviour responses [38,39,40]. The results obtained demonstrate that MEDL was able to attenuate bradykinin-induced nociception, thus, suggesting the extract direct inhibitory effect on the synthesis/action of bradykinin or the B_2_ receptors or indirect action via the modulation of PKC, TRPV1 or glutamate actions that results in the observed antinociceptive activity. Interestingly, the ability of PKC-activating phorbol esters and bradykinin to lower the thermal activation threshold of TRPV1 to below body temperature have also been cited by [37]. Hence, the ability of MEDL to reverse the thermal-induced nociception could plausibly be attributed to the extract ability to inhibit PKC and bradykinin action as seen using the PMA- and bradykinin-induced paw licking test, respectively. Although there is lack of report to link the bradykininergic system with opioidergic system in pain transmission, reports have shown that dynorphin A, an endogenous opioid peptide, induces calcium influx via voltage-sensitive calcium channels in sensory neurons by activating bradykinin receptors to exert the pronociceptive effect [41]. Based on the brief mechanism of action of dynorphin A as described above, it is plausible to suggest that MEDL, being an opioid-acting agent, acts in a contradicting manner on the bradykinin receptors to exert the antinociceptive activity.

The involvement of PKC in the modulation of antinociceptive activity exerted by MEDL was determined using the PMA-induced paw licking test. PKC is a family of serine/threonine kinases that have been found to localize in the anatomical regions that regulate pain and has been demonstrated to play important roles in various intracellular events including pain modulation and analgesia [42]. Activation of PKC, in particular, increases depolarization of TRPV1 receptors, thus, reduces the nociceptive threshold. Interestingly, several inflammatory mediators, including bradykinin and glutamate, may augment the activity of TRPV1 via PKC-dependent pathways [43]. The role of PKC in opioid receptor phosphorylation particularly in opioid desensitization and internalization has also been cited by Ueda et al. [44]. Taking into account that MEDL inhibited PMA-induced nociception, it is possible to suggest that MEDL exerts antinociceptive activity partly by inhibiting the phosphorylation of PKC, which might lead to inhibition of TRPVI receptors and, attenuation of bradykinin and/or glutamate effects on nociceptive transmission.

The role of the opioidergic system in the modulation of antinociceptive activity exhibited by MEDL was determined using the abdominal constriction and hot plate tests. Opioid receptors are found in the PNS and throughout the CNS. Opioid drugs have been used for decades for the management of both acute and chronic pain and exert analgesic effects via directly binding to µ-opioid receptors presynaptically and postsynaptically within the dorsal horn of the spinal cord [45,46]. In addition, Alvarez et al. [47] has also cited that the blunt response of the μ-opioid receptor to its agonists is attributed to the translocation and activation of PKC, the induction of nitric oxide synthase, as well as the increase of intracellular nitric oxide resulting from the postsynaptic influx of calcium ions produced by activated NMDA receptor channels. Hence, an inhibition of NMDA-mediated calcium influx by any agent can also prevent inhibitory effects of PKC and intracellular NO on μ-opioid receptor activity. The present study revealed that the MEDL-exerted antinociceptive activity was inhibited by naloxone, a non-selective antagonist of opioid receptors, when assessed using the abdominal constriction and hot plate tests. Thus, these observations confirmed the involvement of opioid receptors in the modulation of MEDL-exerted antinociceptive activity at the peripheral and central levels. Furthermore, the ability of MEDL to exhibit antinociceptive activity via the modulation of opioid receptors, TRPV1 receptors, glutamatergic system, bradykininergic system and PKC pathway as reported here was concurrent with the previous reports on the association between these nociceptive transmission systems as briefly described above. Interestingly, Alvarez et al. [47] have also reported that the μ-opioids can exert acute antinociceptive activity by binding to opioid receptors, and they can also trigger hyperalgesia by acting on NMDA receptors. The fact that MEDL demonstrated a characteristic of an opioid agonist and produced an antinociceptive activity like MOR, but without inducing hyperalgesia, suggest that the extract also possesses a potential to act as the antagonist rather than agonist of NMDA receptors.

The involvement of l-arginine/NO/cGMP pathways in modulating the antinociceptive activity of MEDL was also investigated using the abdominal constriction test. It is well-known that the l-arginine/NO/cGMP pathway plays a role in the nociceptive transmission at various levels of the sensory system [48]. Depending on the dose and site of administration of NO donors or inhibitors, tissue level of NO or predominant type of fibres involved in the nociceptive response stimulation, this pathway may modulate nociceptive or antinociceptive activities [49,50]. Nitric oxide (NO), a soluble gas continuously synthesized from l-arginine in endothelial cells by the nitric oxide synthase (NOS), modulates pain mechanism at both the PNS and CNS levels depending on its dose. At high dose, NO induces pain while at low dose NO triggers antinociceptive effect [50]. The differential effect exerted by NO might be attributed to the presence of different subsets of primary nociceptive neurons that innervates the tissues [51]. Based on the results of the present study, increase in NO level alone (as seen in the group receiving l-arginine alone) did not affect the nociceptive action triggered by acetic acid but reversed the antinociceptive effect of MEDL (as seen in the group receiving l-arginine and then MEDL). Failure of l-arginine alone to affect the nociceptive action could be due to the inadequate dose of NO (20 mg/kg) used, which may result in an insufficient amount of NO synthesized. However, the same amount of NO produced by l-arginine was sufficient enough to reverse but not block the antinociceptive activity of MEDL. This observation is concurrent with the suggestion that: (i) the effect of NO depends on dosage levels and the rate and timing of its release and (ii) NO might act as a mediator or modulator in analgesic drug’s function. On the other hand, reduction in NO level alone (as seen in the group receiving l-NAME alone) induced antinociceptive action against acetic acid-induced nociceptive response but did not affect the extract-induced antinociceptive activity (as seen in the group receiving l-NAME and then MEDL). The presence of antinociceptive activity at a low level of NO suggests the importance of NO in the modulation of nociceptive transmission. Failure of l-NAME to enhance but instead maintain the antinociceptive activity of MEDL might suggest that: (i) the sufficient level of NO reduced by l-NAME at the peripheral level inactivate several NO-associated (such as COX, glutamatergic, or TRPV1 systems), but not all, nociceptive pathways, or (ii) MEDL directly activates the non-NO-associated nociceptive pathway(s). Moreover, the presence of NO seems to be more dominance over the absence of NO (as seen in the group receiving l-arginine and l-NAME) leading to (i) reversal of antinociceptive effect associated with l-NAME administration, and (ii) incomplete inhibition of the antinociceptive activity of MEDL.

NO triggers guanylyl cyclase resulting in the conversion of guanosine triphosphate to the second messenger, cGMP, which activates cGMP-dependent kinases that modulate numerous direct biological actions, related to NO including spinal nociceptive processing [52]. Based on the results obtained, it is plausible to suggest that modulation of cGMP synthesis plays a role in the nociceptive transmission. It was observed that inhibition of guanylyl cyclase action leads to antinociceptive activity (as seen in the group receiving ODQ alone) and when inhibited in the presence of MEDL (as seen in the group receiving ODQ and then MEDL) no changes in the effectiveness of antinociceptive activity of MEDL was observed. This finding indicates that inhibition of guanylyl cyclase, which leads to blocking of cGMP synthesis, might contribute to the antinociceptive effect. However, the inhibition of guanylyl cyclase neither helps to enhance nor play a role in the antinociceptive activity of MEDL. This seems to suggest that MEDL works via the cGMP-independent pathway. Increase in the level of NO in a condition wherein the action of guanylyl cyclase was inhibited results in: (i) antinociceptive effect associated with ODQ administration (as seen in group receiving l-arginine and ODQ alone), which seems to suggest that the inhibition of guanylyl cyclase whether in the presence of NO or not may lead to antinociceptive activity, and; (ii) failure to neither enhance nor affect the antinociceptive activity of MEDL (as seen in group receiving l-arginine and ODQ and then MEDL). Although the presence of NO has been earlier proven to reverse MEDL activity, inhibition of guanylyl cyclase activity resulted in the failure of NO to affect the antinociceptive activity of MEDL. Based on these observations, it is suggested that the inhibition of cGMP pathway, whether in the presence or absence of NO, will lead to an antinociceptive effect. However, inhibition of cGMP pathway did not cause a significant increase in the antinociceptive activity of MEDL suggesting the involvement of cGMP-independent pathway. This finding is contradicting several reports on the role of cGMP in modulating the antinociceptive activity of several analgesic drugs [53,54].

Nociceptive neuron sensitivity is modulated by a large variety of mediators in the extracellular space. These mediators activate a large number of receptor classes, which in turn activate a plethora of signalling cascades. This might explain the ability of MEDL to affect different nociceptive pathways namely TRPV1 receptors, glutamatergic and bradykininergic system, PKC activity, opioidergic system, and l-arginine/NO/cGMP pathway. Interestingly, different reports have demonstrated the link of l-arginine/NO/cGMP pathway with TRPV1 receptors [55], glutamatergic [56] and bradykininergic [57] system, PKC activity [58], and opioidergic system [59]. How this multitude of cascades mediates nociceptor sensitization and pain is only beginning to be understood.

Plant-derived phytoconstituents have been reported to play role in modulating various pharmacological activity, including antinociceptive activity. As part of an attempt to establish the antinociceptive potential of *D. linearis* and to promote the use of medicinal plants as pain-relieving agent, MEDL was also subjected to phytoconstituents analyses using the UHPLC-ESI-HRMS and GC-MS methods to determine the presence of polyphenolics or any volatile bioactive compounds with potential antinociceptive activity, respectively. The UHPLC-ESI-HRMS analysis of MEDL leads to identification of approximately 30 polyphenolic compounds of which several of them, such as gallic acid [60,61], ferulic acid [62], protocatechuic acid [63], caffeic acid [64,65], *p*-coumaric acid [66], rutin [67,68], isoquercitrin [69], astragalin [70], catechin [71], quercetin [72,73], apigenin [74] and kaempferol [75], have been reported to show antinociceptive activity. These reports also revealed that: (i) gallic acid was reported to show low antinociceptive activity against the acetic acid-induced nociception [60] while its derivative (gallic acid ethyl ester) was reported to attenuate bradykinin- and formalin-induced nociception [61]. In addition, gallic acid ethyl ester was ineffective in the hot-plate test and demonstrated partly the opioid/NO-independent action; (ii) ferulic acid exerts an opioid-mediated antinociceptive activity when assessed using the hot plate test [62]; (iii) protocatechuic acid also exerts an opioid-mediated antinociceptive activity when assessed using the hot plate test [63]; (iv) caffeic acid demonstrated the antinociceptive activity against the abdominal constriction test and the late phase of the formalin-induced nociception, but not the hot plate test [64] whereas the dodecyl ester derivative of caffeic acid were reported to produce antinociceptive activity against the abdominal constriction test as well as the formalin-, capsaicin- and glutamate-induced nociceptive model [65]; (v) rutin exerts antinociceptive activity when assessed using the abdominal constriction test and the formalin-induced nociception, respectively [66,67] with Hernandez-Leon et al. [67] also showed that rutin produces an opioid-mediated antinociceptive activity only in the late phase of the formalin-induced test; (vi) isoquercitrin exhibits antinociceptive activity against the abdominal constriction and formalin tests [68]; (vii) astragalin demonstrates an opioid-mediated antinociceptive activity when assessed using the hot plate test and the formalin-induced nociception [69]; (viii) catechin produces antinociceptive activity against the abdominal constriction, hot plate and formalin-induced paw licking tests with an opioid-independent activity shown using the hot plate test [70]; (ix) quercetin was earlier reported to show an opioid-mediated antinociceptive activity when assessed using the hot plate test [71] while later study demonstrates that quercetin produces antinociceptive activity against the abdominal constriction test, as well as the formalin-, capsaicin- and glutamate-induced nociceptive tests that involves an interaction with l-arginine/NO pathway [72]; (x) apigenin was found to show antinociceptive activity against the abdominal constriction, hot plate and formalin-induced paw licking tests with the centrally-mediated opioid activity proven using the hot plate test [73]; and (xi) kaempferol derivatives (i.e., kaempferol 3-*O*-rutinoside, kaempferol 3-*O*-glucoside and kaempferol-3,7-di-*O*-α-l-rhamnopyranoside) was reported to exert antinociceptive activity against the abdominal constriction and formalin-induced paw licking tests [74,75]. In addition, Ali et al. [75] also reported that kaempferol-3,7-di-*O*-α-l-rhamnopyranoside exerts an opioid-independent antinociceptive activity. Further analysis of MEDL using the GCMS leads to the identification of three volatile bioactive compounds that possessed antinociceptive activity, namely 9,12,15-octadecatrienoic acid, hexadecanoic acid and linoleic acid [76,77,78]. Although some of the findings related to the individual bioactive compound cited above contradicted the present finding related to MEDL, particularly on the involvement of opioid-independent antinociceptive activity, the discrepancy could be attributed to the synergistic action between all of the compounds present in MEDL. The interaction between these compounds is believed to overcome the individual compound effect, which might explain why MEDL exert an opioid-mediated antinociceptive activity.

## 5. Conclusions

In conclusion, MEDL exerts antinociceptive activity at the peripheral and central level via mechanisms of action that involved partly modulation of the TRPV1 and opioid receptors, glutamatergic and bradykininergic system, PKC activity and l-arginine/NO-dependent, cGMP-independent pathway. The antinociceptive activity of MEDL could be due to the presence of several volatile and non-volatile bioactive compounds that have previously proven to attenuate nociceptive response in rodents.

## Figures and Tables

**Figure 1 biomolecules-10-00280-f001:**
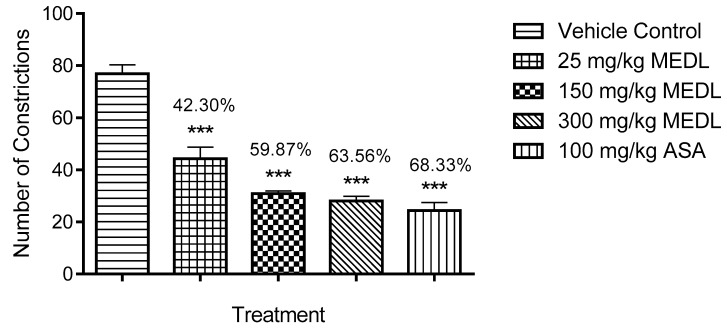
Effects of MEDL on acetic acid-induced abdominal constriction test in mice. Each column represents the mean ± SEM of six mice. Statistical analyses were performed using 1-way ANOVA followed by Tukey’s multiple comparisons test. *** *p* ≤ 0.001 compared to vehicle control group. Values on top of each column denote inhibition.

**Figure 2 biomolecules-10-00280-f002:**
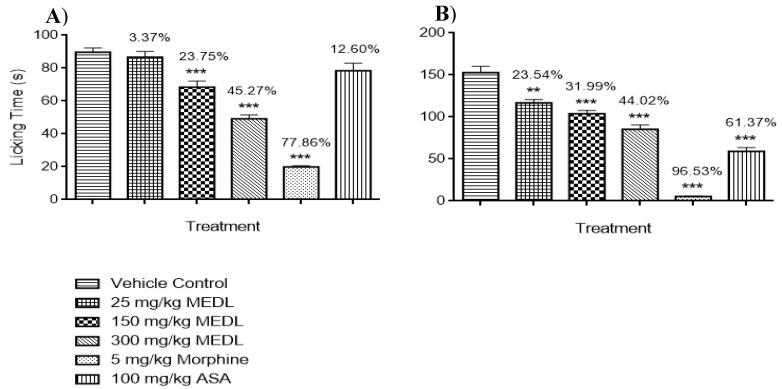
Effects of MEDL on formalin–induced paw licking test in mice. (**A**) Early phase; (**B**) Late phase. Each column represents the mean ± SEM of six mice. Statistical analyses were performed using 1-way ANOVA followed by Tukey’s multiple comparisons test. *** *p* ≤ 0.001 and ** *p* ≤ 0.01 compared to vehicle control group. Values on top of each column denote inhibition.

**Figure 3 biomolecules-10-00280-f003:**
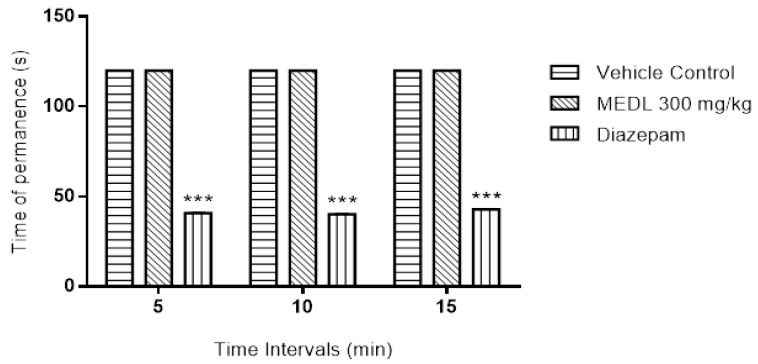
Effect of MEDL on Rota–rod test in mice. Each column represents the mean ± SEM of six mice. Statistical analyses were performed using 2-way ANOVA followed by Tukey’s multiple comparisons test. *** *p* < 0.001 compared to vehicle control group.

**Figure 4 biomolecules-10-00280-f004:**
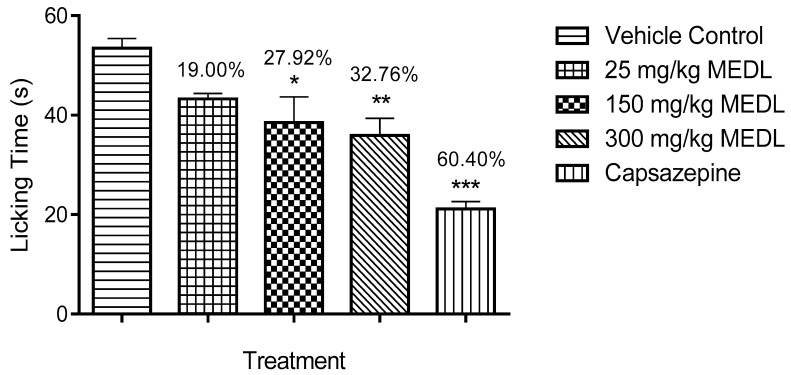
Effect of MEDL on capsaicin-induced nociception in mice. Each column represents the mean ± SEM of six mice. Statistical analyses were performed using 1-way ANOVA followed by Tukey’s multiple comparisons test. * *p* ≤ 0.05, ** *p* ≤ 0.01 and *** *p* ≤ 0.001 compared to vehicle control group. Values on top of each column denote inhibition.

**Figure 5 biomolecules-10-00280-f005:**
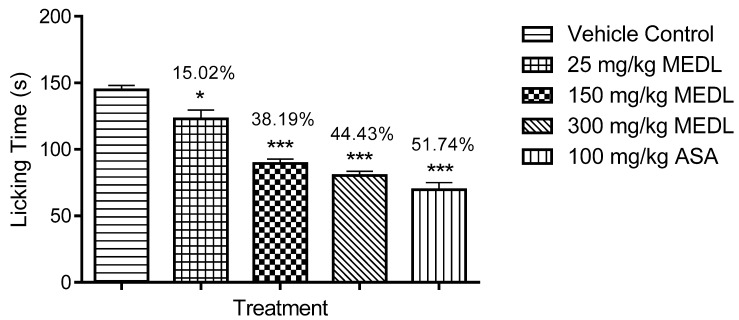
Effect of MEDL on glutamate-induced nociception in mice. Each column represents the mean ± SEM of six mice. Statistical analyses were performed using 1-way ANOVA followed by Tukey’s multiple comparisons test. * *p* ≤ 0.05 and *** *p* ≤ 0.001 compared to vehicle control group. Values on top of each column denote inhibition.

**Figure 6 biomolecules-10-00280-f006:**
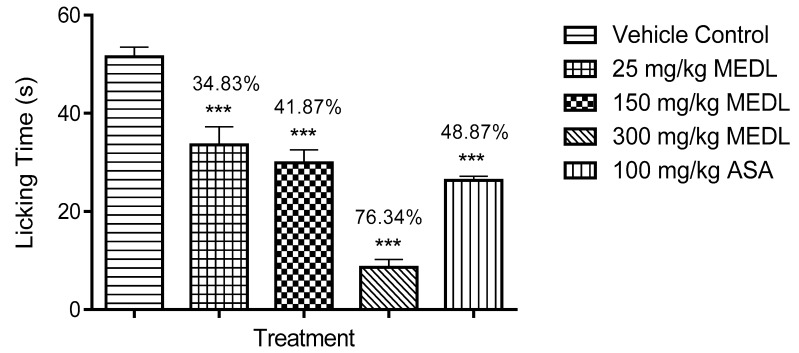
Effect of MEDL on bradykinin-induced nociception in mice. Each column represents the mean ± SEM of six mice. Statistical analyses were performed using 1-way ANOVA followed by Tukey’s multiple comparisons test. *** *p* ≤ 0.001 compared to vehicle control group. Values on top of each column denote inhibition.

**Figure 7 biomolecules-10-00280-f007:**
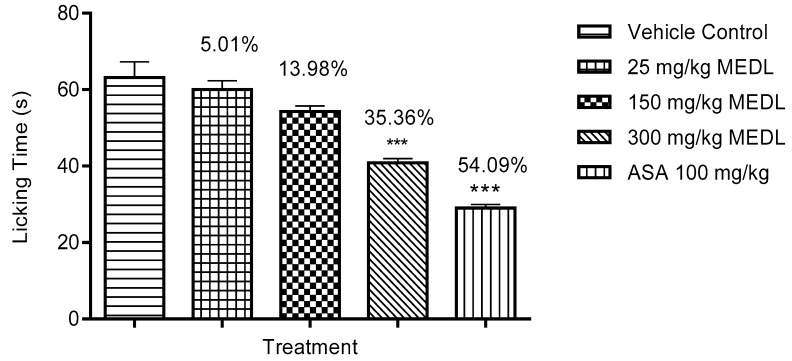
Effect of MEDL on PMA-induced nociception in mice. Each column represents the mean ± SEM of six mice. Statistical analyses were performed using 1–way ANOVA followed Tukey’s multiple comparisons test. *** *p* < 0.001 compared to vehicle control group. Values on top of columns denote inhibition.

**Figure 8 biomolecules-10-00280-f008:**
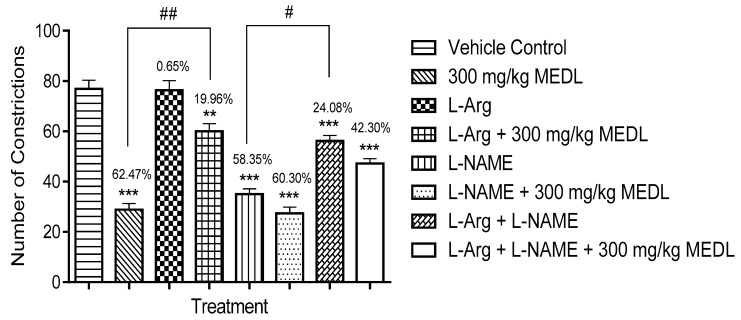
Effects of l-arg or l-NAME on MEDL against acetic acid-induced abdominal constriction test in mice. Each column represents the mean ± SEM of six mice. Statistical analyses were performed using 1-way ANOVA followed by Tukey’s multiple comparisons test. ** *p* ≤ 0.01 and *** *p* ≤ 0.001 compared to vehicle control group; ^#^
*p* < 0.05 and ^##^
*p* < 0.01 compared to 300 mg/kg MEDL or l-NAME-treated group. Values on top of each column denote inhibition.

**Figure 9 biomolecules-10-00280-f009:**
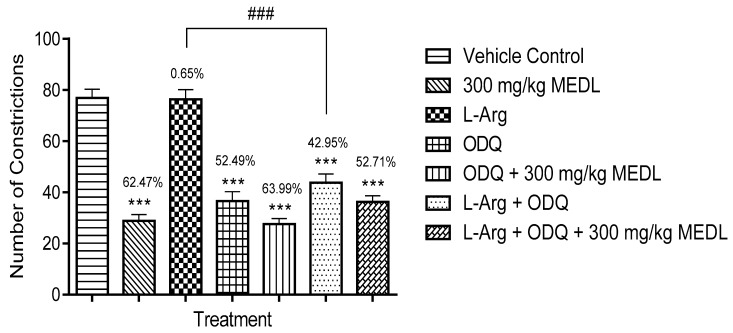
Effects of ODQ on MEDL against acetic acid–induced abdominal constriction test in mice. Each column represents the mean ± SEM of six mice. Statistical analyses were performed using 1-way ANOVA followed by Tukey’s multiple comparisons test. *** *p* ≤ 0.001 compared to vehicle control group; ^###^
*p* ≤ 0.001 compared to l-arginine-treated group. Values on top of each column denote inhibition.

**Figure 10 biomolecules-10-00280-f010:**
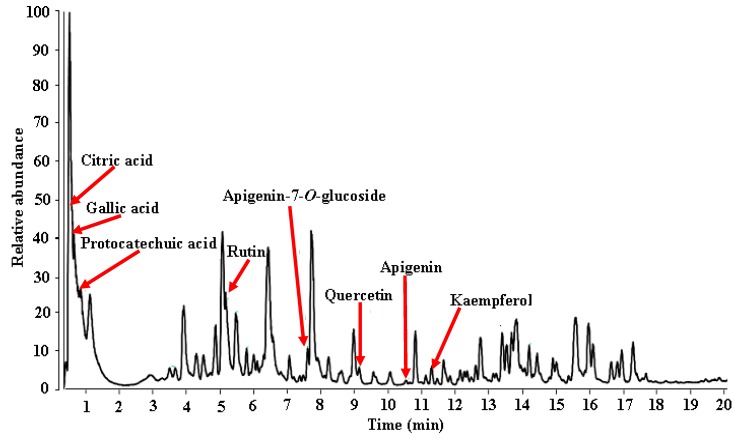
LCMS-Orbitrap fingerprinting analysis of MEDL in negative ionization.

**Figure 11 biomolecules-10-00280-f011:**
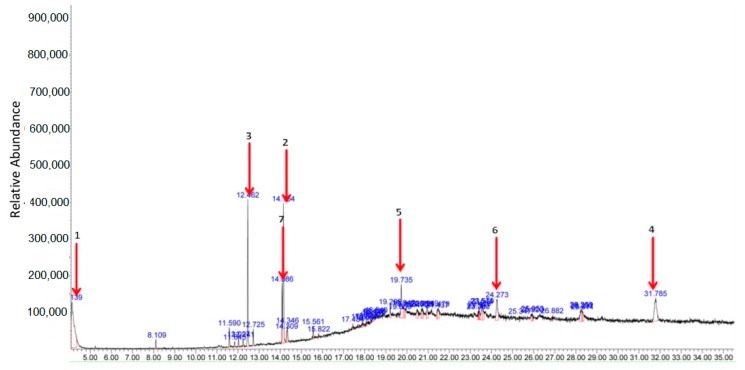
GC-MS profile of MEDL showing 48 detected peaks with 7 major peaks of area more than 4%. (i) Triphenylphosphine oxide (17.52%), (ii) 9,12,15-octadecatrienoic acid (13.43%), (iii) hexadecanoic acid (9.70%), (iv) tri(2-ethylhexyl) trimellitate (7.98%), (v) erucylamide (5.45%), (vi) 5,10-dihexyl-5,10-diihydroindolo[3,2-b]indole-2,7-dicarbaldehyde (4.63%) and (vii) linoleic acid (4.17%).

**Table 1 biomolecules-10-00280-t001:** Effects of MEDL on the hot plate test in mice. Mice were treated with vehicle (10 mL/kg, p.o.), MEDL (25, 150, 300 mg/kg, p.o.), or MOR (5 mg/kg, p.o.).

Treatment	Dose (mg/kg)	Latency of Discomfort (sec) at Respective Time Interval (min)						
		0	60	90	120	150	180	210
Vehicle control		5.53 ± 0.22	5.86 ± 0.19	5.90 ± 0.09	5.50 ± 0.11	5.80 ± 0.25	5.62 ± 0.17	5.65 ± 0.10
MEDL	25	6.14 ± 0.16	7.17 ± 0.14	6.47 ± 0.21	6.77 ± 0.07	6.38 ± 0.22	6.30 ± 0.18	6.25 ± 0.15
	150	6.30 ± 0.30	9.02 ± 0.46 ***	6.93 ± 0.38	6.55 ± 0.13	6.20 ± 0.26	6.05 ± 0.38	6.18 ± 0.36
	300	5.92 ± 0.18	10.83 ± 0.74 ***	8.34 ± 0.25 ***	8.02 ± 0.31 ***	6.83 ± 0.26	6.43 ± 0.29	6.57 ± 0.43
MOR	5	5.80 ± 0.22	12.07 ± 0.90 ***	10.29 ± 0.39 ***	9.14 ± 0.36 ***	8.59 ± 0.43 ***	7.69 ± 0.21 ***	7.65 ± 0.31 ***

Data expressed are the mean ± SEM of reaction time (sec) of six mice. Statistical analysis was performed using 2–way ANOVA followed by Tukey’s multiple comparisons test. *** *p* ≤ 0.001 compared to vehicle control group.

**Table 2 biomolecules-10-00280-t002:** Effects of naloxone on MEDL–induced antinociception in the hot plate test in mice.

Group	Dose (mg/kg)	Latency of Discomfort (sec) at Respective Time Interval (min)						
		0 min	60 min	90 min	120 min	150 min	180 min	210 min
Vehicle Control		5.53 ± 0.22	5.86 ± 0.19	5.90 ± 0.09	5.50 ± 0.11	5.80 ± 0.25	5.62 ± 0.17	5.65 ± 0.10
MEDL	300	5.92 ± 0.18	10.83 ± 0.74 ***	8.34 ± 0.25 ***	8.02 ± 0.31 ***	6.83 ± 0.26	6.43 ± 0.29	6.57 ± 0.43
NLX + MEDL	5 + 300	6.12 ± 0.22	6.21 ± 0.22 ^###^	6.30 ± 0.09 ^###^	5.88 ± 0.30 ^###^	6.03 ± 0.36	6.17 ± 0.27	5.70 ± 0.28
MOR	5	5.80 ± 0.22	12.07 ± 0.90 ***	10.29 ± 0.39 ***	9.14 ± 0.36 ***	8.59 ± 0.43 ***	7.69 ± 0.21 ***	7.65 ± 0.31 ***
NLX + MOR	5 + 5	6.58 ± 0.24	7.08 ± 0.24 ^###^	7.40 ± 0.21 ^###^	7.58 ± 0.40 ^#^	7.03 ± 0.36 #	7.33 ± 0.40	7.23 ± 0.40

Data expressed are the mean ± SEM of latency time (seconds) of six mice. Statistical analyses were performed using 2-way ANOVA followed by the Tukey’s multiple comparisons test. *** *p* ≤ 0.001 compared to vehicle control group; ^#^
*p* ≤ 0.05 and ^###^
*p* ≤ 0.001 compared to 300 mg/kg MEDL or MOR-treated group.

**Table 3 biomolecules-10-00280-t003:** Phytochemical compounds detected and characterized in MEDL using UHPLC-ESI-HRMS in negative ion mode.

Tentative Identification	R_T_ (min)	Mol Formula	Exact Mass [M − H]^−^	∆ Mass (ppm)	MS^n^
Quinic acid	0.6	C_7_H_11_O_6_	191.05495	−0.338	0
Citric acid	0.62	C_6_H_7_O_7_	191.01857	−0.309	11,117,367
Gallic acid	0.65	C_10_H_9_O_4_	169.0.1297	−1.064	12,512,796
Ferulic acid	0.69	C_7_H_5_O_6_	193.04971	0.905	87,111,178,134
Protocatechuic acid	0.87	C_7_H_5_O_4_	153.18720	3.169	109
Protocatechuic acid-4-*O*-β-hexoside	1.1	C_13_H_15_O_9_	315.07184	2.481	0
Coumaryl-hexoside	2.89	C_15_H_17_O_8_	325.09268	2.633	145,163,119
Ferulic acid hexose	3.50	C_9_H_7_O_4_	355.1033	2.651	193,178,134
Caffeic acid	3.52	C_9_H_7_O_4_	179.03387	−0.085	135,107
Liquiritin-*O*-glucosylapioside	3.53	C_16_H_19_O_10_	711.2146	2.122	433,311
Galloylquinic acid	3.62	C_9_H_7_O_3_	343.06738	4.101	0
Quercetin-*O*-diglucoside	4.28	C_27_H_29_O_17_	625.14069	1.223	463,301,151
*p*-Coumaric acid	4.58	C_14_H_15_O_11_	163.03922	1.530	
Rutin isomer ii	4.86	C_27_H_29_O_16_	609.14557	0.918	300,301,151
Dichotomain B-i	5.05	C_21_H_23_O_12_	467.11954	2.435	112
Isoquercetrin	5.53	C_21_H_19_O_12_	463.08820	2.370	300,301,178,151
Dichotamain B-i	5.82	C_21_H_23_O_12_	467.11945	2.242	112
Vicenin	6.1	C_27_H_29_O_15_	593.1033	1.321	575,473,353
Kaempferol-3-*O*-galactoside	6.39	C_21_H1_9_O_11_	447.09283	1.436	284,255
Dichotomain A-i	6.45	C_23_H_25_O_13_	509.12955	1.145	474
Astragalin	6.48	C_21_H_19_O_11_	447.09296	1.727	28,785
(+)Aromadendrin	6.82	C_18_H_17_O_9_	287.05521	0.681	0
Geshoidin	7.06	C_15_H_11_O_7_	377.08701	0.800	217,115
Catechin	7.27	C_15_H_13_O_6_	289.0712	1.852	87,245
Apigenin-7-*O*-glucoside	7.59	C_21_H_19_O_9_	431.09763	0.828	28,587
Dichotomain A-ii	7.71	C_23_H_25_O_13_	509.12955	0.477	474
Quercetin	9.11	C_15_H_9_O_7_	301.03580	1.000	178,151
Kaempferol-3-*O*-glucoside	9.62	C_30_H_25_O_13_	593.12970	1.236	285,161
Apigenin	10.7	C_15_H_9_O5	269.04559	0.521	151,225,228
Kaempferol	11.17	C_15_H_9_O_6_	285.0394	0.125	183,257

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
