# Peer review of "Methanol Extract of Dicranopteris linearis Leaves Attenuate Pain via the Modulation of Opioid/NO-Mediated Pathway"

_biomolecules, 2020, doi:10.3390/biom10020280_

Round 1

Reviewer 1 Report

It contains several valuable data but needs to improve.

1) Please reorganize results with Figures with logics. Current design and flow of results presentation are not satisfied. To do this, please read other papers about anti-nociceptive activity of natural products.

2) Some title of results is unusual. Please rewrite according to other papers about anti-nociceptive research articles.

3) Author suggested that several pathways are involved in activity of MEDL. Please discuss with relationship or orders of these pathways.

4) Please discuss the major constituents contributing the anti-nociceptive activity of MEDL.

5) If possible, some cell or in vitro study might be helpful to prove involvement of suggested mechanisms.

Author Response

Journal: Biomolecules (ISSN 2218-273X)

Manuscript ID: biomolecules-593916

Title: Methanol extract of Dicranopteris linearis leaves attenuate pain via the modulation of opioid/NO-mediated pathway

Authors: Zainul Amiruddin Zakaria *, Rushdudin Al Jofri Rosli, Najihah Hanisah Marmaya, Maizatul Hasyima Omar, Rusliza Basir, Muhammad Nazrul Somchit

Rebuttals to Comments by Reviewer 1:

Comments and Suggestions for Authors

It contains several valuable data but needs to improve.

1) Please reorganize results with Figures with logics. Current design and flow of results presentation are not satisfied. To do this, please read other papers about anti-nociceptive activity of natural products.

Respond:

The authors have reorganized the Results as well as the Discussion sections as requested. Basically, the authors have shifted the phytocontituents results to the end paragraph of the Results section. Arrangement was also made in the Materials and Method section, and Discussion section to synchronize the style of presentation.

2) Some title of results is unusual. Please rewrite according to other papers about anti-nociceptive research articles.

Respond:

The authors have re-written some of the title of results as requested.

3) Author suggested that several pathways are involved in activity of MEDL. Please discuss with relationship or orders of these pathways.

Respond:

The authors have revised the Discussion section particularly the section related to the mechanisms of antinociception of MEDL. Additional information to support the relationship between those pathways were presented wherever suitable. Please see Page 17-21 (Line 479-660).

4) Please discuss the major constituents contributing the anti-nociceptive activity of MEDL.

Respond:

The authors have discussed on the major constituents contributing to the antinociceptive activity of MEDL. Please see Page 21-22 (Line 661-707) .

5) If possible, some cell or in vitro study might be helpful to prove involvement of suggested mechanisms.

Respond:

The authors agreed with the reviewer suggestion regarding the use of cell or in vitro study to prove the involvement of the suggested mechanisms and the molecular study using the tissue or in vitro assays are indeed part of the PhD study of one of the co-authors. However, the molecular study using those assays were just commencing in our laboratory and would take some more time before the authors could really get a complete picture on the mechanisms of antinociception of MEDL at in vivo and in vitro levels. Furthermore, the aimed of the present article was to provide the preliminary observations on the possible mechanisms of antinociceptive potential of MEDL using several standard models, which was the first stage of the study, before the authors can proceed to the next stage of the investigation (molecular study). It is impossible to carry out the in vitro study at the moment taking into consideration also the due date for submission to the special issue. Overall, the authors hope the reviewer can consider the restriction that the authors might face if the in vitro study was to be completed and incorporated in this article.

Reviewer 2 Report

The work entitled ‘Methanol extract of Dicranopteris linearis leaves attenuates pain via the modulation of opioid/NO-mediated pathway.’ The authors observed the antinociceptive activities of D. linearis leaves (MEDL) methanol extract using various mouse models. They identified some of the important phytoconstituents in the extract of MEDL MEDLexerts antinociceptive activity via mechanisms of the TRPV1 and opioid receptors, glutamatergic system, PKC activity, and NO/cGMP pathway.

Comments

There is an urgent need to develop analgesics useful for severe pain, such as cancer pain and chronic pain. The discovery of useful analgesics from natural products with few side effects is very useful clinically.

The authors analyzed the methanol extract. They should show why they chose the methanol extract, and they should mention in the discussion about the difference in components from the previously reported cases of chloroform and water extraction. The authors injected MEDL (ex. 25, 150, 300 mg/kg) to the animal model. How did they weigh the amount of crude extract of MEDL? Is only TRIPV1 receptor involved in the onset of pain due to capsaicin? I think that the results are shown in Fig. 10A and B are difficult to conclude because the effect of arginine is unclear. Is there an upper limit on the value (the number of constriction) of acetic acid-induced nociception result? If the vehicle effect is the upper limit, the effect of arginine is not seen.

Minor point

In Table 2, the dose description is missing. In FIG. 5, the legend of a vehicle is not a horizontal stripe There is no description of A and B in Fig10. There is no legend position in Fig10.

Author Response

Journal: Biomolecules (ISSN 2218-273X)

Manuscript ID: biomolecules-593916

Title: Methanol extract of Dicranopteris linearis leaves attenuate pain via the modulation of opioid/NO-mediated pathway

Authors: Zainul Amiruddin Zakaria *, Rushdudin Al Jofri Rosli, Najihah Hanisah Marmaya, Maizatul Hasyima Omar, Rusliza Basir, Muhammad Nazrul Somchit

Rebuttal to Comments by Reviewer 2:

Comments and Suggestions for Authors

The work entitled ‘Methanol extract of Dicranopteris linearis leaves attenuates pain via the modulation of opioid/NO-mediated pathway.’ The authors observed the antinociceptive activities of D. linearis leaves (MEDL) methanol extract using various mouse models. They identified some of the important phytoconstituents in the extract of MEDL. MEDL exerts antinociceptive activity via mechanisms of the TRPV1 and opioid receptors, glutamatergic system, PKC activity, and NO/cGMP pathway.

Comments

There is an urgent need to develop analgesics useful for severe pain, such as cancer pain and chronic pain. The discovery of useful analgesics from natural products with few side effects is very useful clinically.

The authors analyzed the methanol extract. They should show why they chose the methanol extract, and they should mention in the discussion about the difference in components from the previously reported cases of chloroform and water extraction.

Respond:

The authors have added information on why methanol extract was chosen in the present study in comparison to chloroform and aqueous extracts. A new reference (Reference [22]) was added to support the explanation wherein comparison has been made between the aqueous, chloroform and methanol extract of D. linearis with regard to their phytochemical content, total phenolic content and antioxidant intensity. Justification on using the methanol extract, which was based on the presence of flavonoids, tannins and saponins, high total phenolic content, and high antioxidant activity, was given in the text. Please see Page 14 (Line 402-411).

The authors injected MEDL (ex. 25, 150, 300 mg/kg) to the animal model. How did they weigh the amount of crude extract of MEDL?

Respond:

The dose of MEDL was chosen based on the suggestion made by Schmeda-Hirschmann and Yesilada (2005), who suggested the dose of crude extract for animal study (in this case, gastroprotective activity) to be in the range of 100 to 300 mg/kg. Based on the standardize volume, which has been accepted to be 10 ml/kg, the following step shows how the chosen dose was calculated.

Dose of 300 mg/kg

300 mg in 1000 g animal. Therefore, for a 30 g mice, 9 mg of crude MEDL was required. Since 6 mice were used per group (n=6), 9 mg x 10 mice = 90 mg of crude MDEL was weighed. Based on the standardized volume to be administered to mice that has been set at 10 ml/kg, 10 ml in 1000 g animal or 0.3 ml in 30 g mice will be used. For 10 mice, 3.0 ml volume of vehicle was prepared. Lastly, 90 mg of crude MEDL (From step 2) was dissolved in 3.0 ml vehicle (From step 3). The prepared mixture was then ready to be used in the experiment.

Is only TRIPV1 receptor involved in the onset of pain due to capsaicin?

Respond:

Based on our literature search, yes capsaicin elicits burning pain by activating specific (TRPV1) receptors on sensory nerve endings. However, it was the TRPV1 receptor that can be activated by various stimulus. However, it is safe to say that actually the capsaicin-induced nociceptive model has been widely used to study the role of TRPV1 receptor in the modulation of antinociceptive activity of a compound.

I think that the results are shown in Fig. 10A and B are difficult to conclude because the effect of arginine is unclear. Is there an upper limit on the value (the number of constriction) of acetic acid-induced nociception result? If the vehicle effect is the upper limit, the effect of arginine is not seen.

Respond:

The dose of L-arginine used in the present study was based on earlier report by Dambisya and Lee (1995) and Abacıoglu et al. (2000). Dambisya and Lee (1995) have reported that, at 20 mg/kg, L-arginine had no effect on the number of abdominal constrictions induced by acetic acid. Contradictory to this finding, was the report made by Abacıoglu et al. (2000), who reported that 20 mg/kg L-arginine significantly increased the number of abdominal constriction (induced nociception). Our earlier investigation using the same concentration of L-arginine revealed that L-arginine did not cause any significant changes in the nociceptive response induced by acetic acid. It is therefore postulated that the concentration of NO, produced from the conversion of 20 mg/kg L-arginine in the body, together with the endogenously available NO were not enough to enhance the nociceptive response induced by acetic acid. However, the fact that 20 mg/kg L-arginine significantly reversed the antinociceptive activity of MEDL did indicate the involvement of NO in the modulation of MEDL-induced antinociceptive activity. In comparison, Dambisya and Lee (1995) have also reported that 20 mg/kg L-arginine failed to affect morphine-induced antinociceptive activity, which suggested that morphine might possibly works via a NO-independent pathway. Interestingly, the present study shows that the L-arginine reversed the antinociceptive activity of MEDL. Based on the above reports, the same dose of L-arginine (20 mg/kg) was used in various antinociceptive study (Sani et al., 2012; Zakaria et al. 2014, Abdul Rahim et al., 2016).

References:

Dambisya YM, Lee TL. Effects L-NG-nitro arginine methyl ester (L-NAME), L-NG-monomethyl arginine (L-NMMA) and L-arginine on the antinociceptive effects of morphine in mice. Methods Find Exp Clin Pharmacol. 1995 Nov;17(9):577-82.

  Abacioğlu N, Tunçtan B, Akbulut E, Cakici I. Participation of the components of L-arginine/nitric oxide/cGMP cascade by chemically-induced abdominal constriction in the mouse. Life Sci. 2000;67(10):1127-37.

Sani MH, Zakaria ZA, Balan T, Teh LK, Salleh MZ. Antinociceptive Activity of Methanol Extract of Muntingia calabura Leaves and the Mechanisms of Action Involved. Evid Based Complement Alternat Med. 2012;2012:890361. doi:10.1155/2012/890361

Zakaria ZA, Jaios ES, Omar MH, et al. Antinociception of petroleum ether fraction derived from crude methanol extract of Melastoma malabathricum leaves and its possible mechanisms of action in animal models. BMC Complement Altern Med. 2016;16(1):488. Published 2016 Nov 29. doi:10.1186/s12906-016-1478-1

Abdul Rahim MH, Zakaria ZA, Mohd Sani MH, Omar MH, Yakob Y, Cheema MS, Ching SM, Ahmad Z, Abdul Kadir A. Methanolic Extract of Clinacanthus nutans Exerts Antinociceptive Activity via the Opioid/Nitric Oxide-Mediated, but cGMP-Independent, Pathways. Evid Based Complement Alternat Med. 2016;2016:1494981. doi: 10.1155/2016/1494981. 

Minor point

In Table 2, the dose description is missing.

Respond:

The authors have added the dose description in Table 2.

In FIG. 5, the legend of a vehicle is not a horizontal stripe.

Respond:

Amendment has been made to the legend of a vehicle in Figure 5.

There is no description of A and B in Fig10.

Respond:

The authors have rearranged the Methodology, Results and Discussion sections as requested by another reviewer due to issue of logics. Due to this, Figure 10A and 10B have been renumbered as Figure 8 and 9, respectively.

There is no legend position in Fig10. 

Respond:

The authors have rearranged the legend for Figure 8 (previously Figure 10A) and Figure 9 (previously Figure 10B) according to the format of the journal.

Round 2

Reviewer 1 Report

OK! Several issues are cleared.

Reviewer 2 Report

Authors answered all my review comments.

The authors added new references and described the reason why they chose methanol extraction. 

The authors showed me how to calculate the dose. 

I agree the capsaicin elicits burning pain by activating specific (TRPV1) receptors on sensory nerve endings.

And they explained about L-arginine dose.